# Food Fortification of Instant Pulse Porridge Powder with Improved Iron and Zinc Bioaccessibility Using Roselle Calyx

**DOI:** 10.3390/nu14194070

**Published:** 2022-09-30

**Authors:** Sandhya Subedi, Uthaiwan Suttisansanee, Aikkarach Kettawan, Chaowanee Chupeerach, Chanakan Khemthong, Sirinapa Thangsiri, Nattira On-nom

**Affiliations:** Food and Nutrition Academic and Research Cluster, Institute of Nutrition, Mahidol University, Salaya, Phuttamonthon, Nakhon Pathom 73170, Thailand

**Keywords:** *Setaria italica* L., *Cicer arietinum* L., *Hibiscus sabdariffa* L., chickpea flour, foxtail millet flour, minerals, organic acids, pregelatinization, sensory evaluation, nutritional values, in vitro bioaccessibility

## Abstract

Undernutrition and mineral deficiencies negatively impact both the health and academic performance of school children, while diets high in phytic acid and some phenolics inhibit the absorption of minerals such as iron and zinc. This study developed instant porridge powders rich in iron and zinc using pregelatinized chickpea flour (PCPF) and pregelatinized foxtail millet flour (PFMF) and assessed the potential of utilizing roselle calyx powder (RCP) as a source of organic acids to enhance its iron and zinc bioaccessibility. Physical properties, nutrients, mineral inhibitors and in vitro iron and zinc bioaccessibility of different proportions of PCPF, PFMF and RCP in instant porridge powders were evaluated. Three instant porridge powder formulations including instant chickpea powder (ICP) using PCPF, instant composite flour (ICF) using PCPF and PFMF and instant pulse porridge powder (IPP) using PCPF, PFMF and RCP were developed. Results show that all instant porridge powders were accepted by sensory evaluation, while different ingredients impacted color, consistency and the viscosity index. Addition of RCP improved the bioaccessibility of iron (1.3–1.6-fold) and zinc (1.3–1.9-fold). A 70 g serving of these instant porridge powders substantially contributed to daily protein, iron and zinc requirement for children aged 7–9 years. These porridge powders hold potential to serve as school meals for young children in low-to-middle income countries.

## 1. Introduction

Iron deficiency anemia is one of the five leading causes of individuals living with disability burden worldwide [1]. Iron deficiency anemia has been widely researched, with the global prevalence of iron deficiency without anemia expected to be at least double that of iron deficiency anemia [2]. Zinc deficiency is also a major public health problem impacting almost all low-to-middle income countries [3]. These deficiencies lead to impaired metal and physical and cognitive development, reduced work productivity, an increased risk of perinatal complications, morbidity from infectious disease and mortality [2,3]. At both individual and population levels, deficiencies of iron and zinc coexist in low-to-middle income countries such as Nepal, Ethiopia and Nigeria. This is because they share similar food sources and dietary elements, hindering or facilitating the absorption of these two minerals [3,4]. The problem is more severe in southern Asia, primarily among women and children, even though this region is witnessing improved agricultural output, poverty alleviation and access to education [4]. Iron and zinc deficiencies arise from the consumption of monotonous, predominantly unrefined plant-based diets. They are usually due to the presence of high amounts of mineral binders such as phytic acid and condensed tannin [5,6]. Wholegrain cereals and legumes may contain sufficient quantities of these minerals to meet daily intake requirements, but their bioavailability is generally low [5].

Food-based strategies include the integration of dietary diversity, with the principles of bioavailability translated to efficient food synergies, and due emphasis on food accessibility and affordability [7]. Such strategies are recognized as cost-effective, viable and sustainable solutions to combat the double burden of malnutrition of at-risk populations, especially school children. They also promote self-reliance of the community [3,7]. The nutritional status of school children has a direct impact on their health, cognitive abilities and educational performance. School children are often far from mainstream beneficiaries in underdeveloped countries such as Nepal, and the available resources mostly go to younger children who are more vulnerable [8]. However, the importance of investment in school children has been recognized to improve nutrition and also derive long-term societal benefits through poverty reduction and diminishing the economic gap [8,9]. A copious amount of evidence suggests that integration of school meal programs to provide complete or supplemental meals to students promotes nutrition, school participation and enrollment and also enhances educational achievement [10,11,12,13]. It is recommended that school meals be prepared using locally available foods high in protein and micronutrients, such as cereal- and legume-based porridge powder [10,11,12,13], which are widely consumed in Nepal by all age groups, particularly children [14].

Foxtail millet (*Setaria italica* L.) is one of the oldest varieties of millet, and a neglected minor crop with great potential to help in coping with food insecurity and climate change in remote areas of Nepal [15]. This gluten-free grain is rich in calcium, dietary fiber, polyphenols and protein [16]. Chickpea (*Cicer arietinum* L.) is one of the major pulses consumed as a reliable source of plant-based protein, vitamins and minerals in Nepal [17]. Utilization of this pulse as a sustainable and ecofriendly source of protein is attracting increased attention, with countless possibilities to improve food insecurity worldwide [18]. To improve released iron and/or zinc contents from the food matrix through digestion and solubility in the gut lumen [19], addition of organic acids, especially from plant sources, is required. Roselle (*Hibiscus sabdariffa* L.) is a locally available phenolics-rich plant containing dietary components such as ascorbic acid, citric acid and other organic acids [20,21]. Combining such locally available plants with staple grains (food-to-food fortification) has been advocated as a sustainable method to improve iron and zinc contents and bioavailability [22]. However, a study of food-to-food fortification of instant porridge powder with improved iron and zinc bioaccessibility using roselle calyx for Nepalese school children is challenging due to the specific characterizations of each ingredient.

Hence, this study developed nutrient-dense instant porridge powder from pregelatinized chickpea flour (PCPF) and pregelatinized foxtail millet flour (PFMF) using roselle calyx powder (RCP) as a potential food-to-food fortification to improve the bioaccessibility of iron and zinc while also maintaining the desired sensory properties. Different proportions of pregelatinized chickpea flour, pregelatinized foxtail millet flour and roselle calyx powder were assessed for their physical properties, nutritive values and mineral inhibitor contents. Results provide crucial information concerning iron and zinc bioaccessibility of roselle calyx-enriched instant porridge powder products, using locally available and inexpensive food sources as possible future school meals.

## 2. Materials and Methods

### 2.1. Raw Materials

Chickpea and foxtail millet were purchased from Young Smart Farmer, Songkhla, Thailand. The pregelatinized forms of these two pulses, PCPF and PFMF, were prepared according to Subedi et al. [23]. PCPF consisted of 4.85% moisture, 22.34% protein, 7.27% total fat, 63.68% total carbohydrates (comprising 17.07% dietary fiber), 4.90% iron and 2.62% zinc, while PFMP consisted of 5.88% moisture, 12.84% protein, 3.58% total fat, 76.60% total carbohydrates (comprising 3.41% dietary fiber), 3.29% iron and 2.52% zinc. The chemical composition of RCP was 9.11% moisture, 7.26% protein, 1.69% total fat, 73.99% total carbohydrates (comprising 41.89% dietary fiber) and 38.60% total organic acids (comprising 18.84% succinic acid, 13.61% formic acid, 2.21% acetic acid, 2.15% oxalic acid, 1.55% citric acid, 0.23% ascorbic acid and 0.01% malic acid). RCP was purchased from Huglamool Farm, Amnat Charoen, Thailand. Dried powders of seasoning herbs (onion powder, garlic powder, cumin seed powder, coriander seed powder, black pepper and turmeric) were bought from Khunsiri brand, Pathum Thani, Thailand. Chili powder was obtained from Prakriti Indian Trading Co., Ltd., Bangkok, Thailand. Soybean oil was purchased from Angoon brand, Thai vegetable oil PLC, Bangkok, Thailand. All reagents used in this study were obtained from Sigma-Aldrich (St. Louis, MO, USA).

### 2.2. Preparation of Three Different Instant Porridge Powder Formulations

Three different formulations of instant porridge powders, including instant chickpea powder (ICP) using PCPF, instant composite flour (ICF) using PCPF and PFMF and instant pulse porridge power (IPP) using PCPF, PFMF and RCP, were prepared (Table 1) by mixing the ingredients with a laboratory blender (model HR1393/00, Philips (Thailand) Ltd., Bangkok, Thailand) for 30 s. The optimal ratio of RCP in the IPP formulation was selected based on sensory evaluation and in vitro bioaccessibility of iron and zinc, as shown in Appendix A. The mixtures were then sealed in aluminum foil bags and kept at 4 °C until use. Seasoning spices mixed powder (SSMP) consisting of 52.83% onion powder, 3.33% garlic powder, 0.67% turmeric powder, 1.67% coriander powder, 0.66% cumin powder, 1.67% chili powder and 0.33% black pepper powder, was prepared using a laboratory blender, packed in aluminum foil bags and kept at 4 °C until use.

### 2.3. Analysis of Nutritive Values

Proximate analysis (moisture content, fat, protein, ash, carbohydrates, energy and total dietary fiber) was determined according to the method of AOAC [24]. Moisture content was evaluated by drying the samples in a hot-air oven (Memmert model UNE 500, Eagle, WI, USA) at 100 °C until constant weight was achieved (AOAC 930.04). Total fat content was determined by acidic digestion and solvent extraction with petroleum ether using a Tecator Soxtec System (model 1043, Hoganas, Sweden) (AOAC 922.06). Protein content was analyzed using the Kjeldahl method utilizing digestion and distillation units (BÜCHI Corporation, New Castle, DE, USA), and then calculated using a conversion factor of 6.25 (AOAC 991.20). Ash content was analyzed by incineration of the organic matter in a muffle furnace (Carbolite model CWF 1100, Hope, UK) at 550 °C (AOAC 930.30). Total carbohydrates were calculated by the subtraction of moisture, fat, protein and ash contents from 100. Energy value was attained from the integration of total energy from carbohydrates, protein and fat as 4, 4 and 9 kcal/g samples, respectively. Total dietary fiber was evaluated using the enzyme gravimetric method (AOAC 985.29). Iron and zinc were analyzed using an Inductively Coupled Plasma–Optical Emission Spectrophotometer (ICP–OES) (AOAC 984.27) [24]. All samples were determined in the Accredited Laboratory, complying with ISO/IEC 17025:2017, at the Institute of Nutrition, Mahidol University (Nakhon Pathom, Thailand). Results as per 100 g fresh weight (FW) are shown in Appendix A. To accurately compare the nutritive values among the three instant porridge powders, compositions per 100 g dry weight (DW) were calculated.

### 2.4. Sensory Evaluation

Sensory evaluation was conducted by 40 untrained panelists comprising Nepali people living in Thailand. The test was performed in an individual testing booth of the sensory science laboratory at the Institute of Nutrition, Mahidol University, and lasted 10–15 min. All samples were prepared on the same day of the test and served fresh. The porridge was prepared by reconstituting porridge powder in warm water (60 ± 2 °C) at a ratio of 1:3 (*w/w*). The panelists evaluated the products for satisfaction based on appearance, color, odor, taste, texture and overall liking using a nine-point hedonic scale (1 = dislike extremely to 9 = like extremely) [25].

### 2.5. Physical Analysis

Physical properties were analyzed in terms of color (L* (lightness), a* (red–green color) and b* (yellow–blue color)) using a Colorflex EZ Spectrophotometer (HunterLab, Reston, VA, USA), water activity (A_w_) by a water activity measurement instrument (model ms1–1 M, Novasina, Lachen, Switzerland) and pH using a pH meter (Ohaus Corporation, Morris Country, NJ, USA). Bulk density (BD) was determined by weighing 50 g of sample into a 100 mL measuring cylinder before tapping continuously until a constant volume was obtained. BD was calculated using Equation (1) [26].
(1)$$\mathrm{Bulk}\;\mathrm{density}\;(g/{\mathrm{cm}}^{3})=\frac{\mathrm{Weight}\;\mathrm{of}\;\mathrm{the}\;\mathrm{sample}}{\mathrm{Volume}\;\mathrm{of}\;\mathrm{the}\;\mathrm{sample}\;\mathrm{after}\;\mathrm{tapping}}$$

The water solubility index (WSI) and water absorption index (WAI) were determined following Anderson, 1999 [27]. The sample (1.25 g) was weighed into a preweighed 50 mL centrifuge tube, followed by the addition of 20 mL of deionized water. The centrifuge tube with the sample mixture was then placed in a SWB20 water bath shaker (Australian Scientific Instruments PTY. Ltd., Fyshwick, Canberra, Australia) at 30 °C for 30 min. The sample was then centrifuged using a Hettich^®^ROTINA 16R centrifuge (Andreas Hettich GmbH, Tuttlingen, Germany) at 1190× *g* for 10 min at 25 °C. The supernatant was decanted for determination of its solid content, and the sediment was weighed using a weighing balance (Mettler Toledo, Toronto, Canada). The supernatant was dried at 105 °C until reaching a stable weight (approximately 3 h). WSI and WAI were calculated using Equations (2) and (3), respectively.
(2)$$\mathrm{WSI}=\left(\frac{\mathrm{Weight}\;\mathrm{of}\;\mathrm{dissolved}\;\mathrm{solid}\;\mathrm{in}\;\mathrm{supernatant}}{\mathrm{Weight}\;\mathrm{of}\;\mathrm{dry}\;\mathrm{solids}}\right)\;\times \;100$$
(3)$$\mathrm{WAI}=\frac{\mathrm{Weight}\;\mathrm{of}\;\mathrm{sediment}}{\mathrm{Volume}\;\mathrm{of}\;\mathrm{dry}\;\mathrm{solid}}$$

The texture of porridge samples as determined by the consistency and viscosity index was analyzed by a back extrusion test using a TA–XT plus Texture Analyzer (Stable Micro Systems, Godalming, Surrey, UK) followed by the method of Syahariza and Yong (2017) [28]. The texture analysis was carried out at 25 °C and replicated ten times for each sample.

### 2.6. Anti-Nutrient Analysis

Phytic acid content (total phosphorus) was measured using a phytic acid assay kit (Megazyme, Wicklow, Ireland) as described by McKie and McCleary (2016) [29], with results expressed as g/100 g DW. To determine total phenol and tannin contents, the Folin–Ciocalteu method of The Food and Agriculture Organization of the United Nations (FAO) and the International Atomic Energy Agency (IAEA) (2000) was utilized [30]. Results are expressed as mg tannic acid equivalent (TAE)/100 g DW.

### 2.7. Determination of Iron and Zinc Bioaccessibility

The samples were analyzed for bioaccessibility by the in vitro dialyzability method of Ting and Loh (2016) [31]. Peptic and pancreatic digestion were simulated by the enzyme pepsin (P7000, from porcine stomach mucosa), pancreatin (P1750, from porcine pancreas) and bile extract (B8631, porcine). Mineral contents of the dialysates were analyzed by ICP-OES as described in the iron and zinc contents analysis.

### 2.8. Microbiological Analysis

Total plate count (TPC) and yeast and mold counts (YMCs) were performed following the standard protocols of the Bacteriological Analytical Manual (BAM) [32]. The numbers of colonies appearing on the dilution plates were counted, averaged and reported as colony forming units (CFUs)/g.

### 2.9. Statistical Analysis

All measurements were performed in triplicate or as stated otherwise, with results expressed as mean ± standard deviation (SD). Mean differences of *p* < 0.05 were determined by one-way analysis of variance (ANOVA), followed by Duncan’s multiple comparison test for more than two data sets or Student’s unpaired *t*-test for two data sets using IBM SPSS Statistics for Windows version 26.0 (IBM Corp., Armonk, NY, USA).

## 3. Results

### 3.1. Nutritional Analysis

The proximate analysis results as well as iron and zinc contents (per 100 g DW) of ICP, ICF and IPP are shown in Table 2, while contents per 100 g FW are shown in Appendix A. The energy, fat, carbohydrates and total dietary fiber of all instant porridge powders varied significantly. ICP exhibited slightly but significantly higher energy (1-fold higher than the others), protein (1.2-fold higher than the others), fat (1.2-fold higher than the others) and total dietary fiber (1.2–1.5-fold higher than the others), while containing the lowest carbohydrate content (1.1-fold lower than the others). The ash content of IPP was also slightly but significantly higher than ICP and ICF (1.1-fold higher), possibly due to the supplementation of RCP. The same trends with ash content were observed in iron contents of all instant porridge powders, while no significant difference in zinc content (2.72–3.95 mg/100 g DW) was observed among the instant porridge powders.

### 3.2. Sensory Evaluation of Instant Porridge Powders

Sensory evaluation results of ICP, ICF and IPP are shown in Table 3, indicating no significant differences (*p* ≥ 0.05) in sensory attribute scores among the three instant porridge powders, including in appearance (6.45–6.73), color (6.68–6.85), odor (6.68–7.10), taste (6.28–6.75), texture (6.20–6.73) and overall liking (6.43–6.85). The sensory attribute scores of all instant porridge powders were higher than 6 (like slightly).

### 3.3. Physical Analysis of Instant Porridge Powders

Physical property results including color, water activity (A_w_), water solubility index (WSI), water absorption index (WAI), bulk density (BD) and texture (consistency and viscosity index) are shown in Table 4. Color analysis indicated that IPP was significantly darker and redder than ICP and ICF, possibly due to the addition of RCP. All instant porridge powders exhibited A_w_ ranging from 0.13 to 0.18, with IPP exhibiting the highest value. No significant differences in WSI (6.13–6.40%), WAI (3.77–4.04 g/g) and BD (0.72 g/cm^3^) were observed among the instant porridge powders. For texture analysis, the highest consistency (1.3–4.3-fold higher than the others) and viscosity index (1.6–25.1-fold higher than others) were recorded in ICF, while ICP exhibited the lowest consistency and viscosity index.

### 3.4. Mineral Inhibitor Contents in Instant Porridge Powders

A plant-based diet constitutes an important source of all nutrients, including minerals. However, plants are also sources of mineral inhibitors such as phytate, the main inhibitor of mineral absorption (iron, zinc and calcium) [33]. Thus, the mineral inhibitor contents of ICP, ICF and IPP, including phytic acid, tannin and phenols, were evaluated (Table 5). Results indicate no significant difference in phytic acid (1.45–1.58 g/100 g DW) among the instant porridge powders. However, the tannin and total phenol contents of all instant porridge powders varied significantly. IPP presented the highest tannin (1.8–3.6-fold higher than the others) and total phenols (1.7–2.5-fold higher than the others), while the lowest tannin and total phenol content were recorded in ICF.

### 3.5. In Vitro Iron and Zinc Bioaccessibility of Instant Porridge Powders

Iron and zinc bioaccessibility and pH values of the three instant porridge powders were determined using the in vitro digestion assay (Table 6). IPP presented the highest bioaccessibility and percentage of bioaccessibility of iron (1.5–1.6-fold and 1.3–1.4-fold, respectively, higher than the others) with zinc (1.9-fold and 1.3–1.5-fold, respectively, higher than the others). However, no significant differences in the bioaccessibility or percentage of iron and zinc bioaccessibility were observed between ICP and ICF. For pH, IPP exhibited the lowest values (1.3-fold lower than the others), while no significant pH difference was recorded between ICP (5.78) and ICF (5.83).

### 3.6. Microbiological Analysis

In this study, the total plate count (TPC) and yeast and mold counts (YMCs) of the instant porridge powders were assessed according to the Thai Community Product Standard based on instant rice porridge powder. This states that the TPC of the product must be less than 1 × 10^6^ CFU/g, while the YMC must not exceed 100 CFU/g [34]. Results show that the TPC and YMC values of the three instant porridge powders were in line with the standard (Table 7).

## 4. Discussion

To combat climate change, promotions of sustainable plant-based food sources are gaining emphasis worldwide. Plant-based food also contains calories and macronutrients. To prevent malnutrition and improve health and cognitive outcomes, the macronutrient balance (ratios between protein, carbohydrates and fat) is a key factor [35,36]. Minerals such as iron and zinc are found abundantly in plant-based foods such as legumes, while mineral inhibitors such as phytate have lower bioavailability compared to animal sources. However, household processing methods such as soaking and addition of enhancers help to ensure that the body uses the minerals we consume. Growing children require higher amounts of these nutrients. Our bodies should be able to easily utilize nutrients to gain the benefits of consuming bioactive compounds from plant-based food sources. This study developed instant porridge powder using pregelatinized foxtail millet and chickpea flours as sources of protein, carbohydrates and fat according to the World Health Organization (WHO) recommendation and fortified the product using roselle calyx powder as a source of organic acids to improve the bioaccessibility of the contained iron and zinc.

The proximate analysis results show that ICP had the highest protein and fat contents owing to having the highest PCPF content. Chickpea is a good source of protein and contains significant amounts of all the essential amino acids. All the instant porridge powders met the recommended values of protein (≥15%) according to FAO and WHO (1991) [37]. The WHO (2009) also suggested that values higher than 15% protein may increase renal solute load and interfere with appetite [38]. The recommended dietary allowances (RDAs) and absolute requirement (AR) of protein in the Indian Council of Medical Research (ICMR) (2020) [39] states that the RDA of protein for children aged 7–9 years is 23 g protein/day. One serving size (70 g) of ICP provided a 1.2-fold higher protein contribution to RDA than ICF and IPP (Table 8). All three instant porridge powders demonstrated a higher content of protein than the RDA, while for fat content, all instant porridge powders met the recommended FAO and WHO (1991) values of fat (10–25%). [37]. The fat content in chickpea varies from 3.80 to 10.20% as a relatively good source of nutritionally important polyunsaturated fatty acids, linoleic acid and monounsaturated oleic acid [40]. The carbohydrate contents of ICF and IPP were also within the recommended levels of FAO/WHO (1991), while ICP did not meet the recommended level (64 ± 4 g/100 g) [37]. This observation suggests that legume and cereal combinations provide more balanced nutrition than a single plant-based food material. The dietary fiber contents of the three instant porridge powders also showed significant differences, with the highest value recorded in ICP. According to the ICMR (2020) recommendation, dietary fiber intake should be 25 g/day [39]. IPP presented the highest dietary fiber (1.2–1.5-fold higher than the others), possibly due to the high proportion of PCPF. All instant porridge powders were high in fiber products, with dietary fiber contents of more than 6 g/100 g [41]. Energy is provided by proteins, fats and carbohydrates. The highest energy was recorded in ICP, due to the relatively high protein and fat contents in PCPF. All instant porridges were energy-dense foods that provide many calories in a small serving amount.

Ash content represents the total mineral content in foods. The three instant porridge powders varied significantly in ash content due to the differences in amounts of individual ingredients. Ash content was highest in IPP, possibly due to the addition of RCP during formulation, as RCP is a food source of minerals [42]. This result was confirmed by the observation that the highest amounts of iron (1.2-fold higher than the others) and zinc (1.1–1.4-fold higher than the others) occurred in IPP. According to the RDA and AR values for iron and zinc in the ICMR (2020) [39], children aged 7–9 years require 15 mg iron/day and 5.90 mg zinc/day. Our results show that IPP had the highest contribution of iron (1.1-fold higher than the others) and zinc (1.3–1.4-fold higher than the other) to the RDA for children aged 7–9 years (Table 8), possibly due to the RCP supplemented in the formulation. The AR of iron (requirement for growth + basal losses + menstrual losses) was based on the WHO/UNICEF/UNU 2004 (95th percentile) for children aged 7–10 years [43]. They reported that the AR of iron for children aged 7–10 years is 0.89 mg/day. The highest contribution to the AR of iron was recorded in IPP (40.10%), followed by ICF (25.96%) and ICP (25.17%) (Table 8). The International Zinc Nutrition Consultative Group (IZiNCG) (2004) set the AR of zinc for children aged 4–8 years with a reference body weight of 21 kg as 0.83 mg/day and the AR of zinc for children aged 9–13 years with a reference body weight of 38 kg as 4.53 mg/day [44]. Total endogenous zinc losses were calculated as 0.034 mg/kg/day for children one year of age and older (i.e., urinary losses, surface losses and intestinal losses), based on their respective reference body weights and rates of weight gain [44]. IPP showed the highest AR of zinc for both children aged 4–8 years with a reference body weight of 21 kg and children aged 9–13 years with a reference body weight of 38 kg (1.9–2.0-fold higher than the others), while ICP had the lowest contribution to the AR of zinc in both age groups. A food is considered to be a good source of nutrients if it fulfills more than 20% of the RDA [39]. All instant porridges were excellent sources of protein, iron and zinc, and could benefit children who suffer from macronutrient (protein) and micronutrient (mineral) deficiency.

The highest moisture content was also found in IPP, and attributed to the high moisture content of the RCP added in this formulation. Moisture content is important to maintain food quality. High moisture content of food results in microbial growth and ultimately destroys quality. El Wakeel (2007) stated that dried materials with lower than 10% moisture content, such as instant soup ingredients, had better storage qualities [45]. All instant porridge powders were deemed microbiologically safe to consume according to the criteria of the Thai community product standard based on instant rice porridge powder [34].

As final products, all instant porridge powders were accepted by the consumers and received sensory attribute scores above 6, or ‘like slightly’. Previous research used an average value of 6 (like slightly) on a nine-point hedonic scale as the minimum acceptable limit for consumers liking a product [46,47,48]. Therefore, ICP, ICF and IPP showed promise to be developed with consumer acceptance. The hedonic rating obtained for IPP was also higher than that for ICP and ICF. Results suggest that the addition of RCP did not adversely affect product sensory perception.

To determine the physical properties, the color of the instant porridge powders was measured. IPP was darker and redder than ICP and ICF, probably due to the intense red pigment of anthocyanins in RCP [49]. Anthocyanins are water-soluble pigments belonging to the phenolic group. They are responsible for red, purple and blue colors in fruits and vegetables [50]. Similar results were reported by Beswa and Singo (2019) [51], who revealed that a high concentration of roselle extracts increased the redness of ice cream, while lightness decreased. Color changes in instant porridge powders occurred because of the unique colors of the ingredients.

Water activity of ICP, ICF and IPP was low (0.13–0.18), and less than the required water activity level for microbe growth [52]. The water solubility index (WSI) and water absorption index (WAI) are parameters used to explain the hydration properties of porridge. The WSI determines the amount of soluble degradation from starch upon the addition of excess of water [27,53] and also indicates the water penetration into starch granules [54]. A lower WSI indicates minor degradation of starch [55]. The WSI values of instant porridge powders in this study, at 6.13–6.40%, were lower than reported those by Mahgoub et al. (2020) [55]. They found that WSI values of instant porridge supplemented with different types of mung bean were 16.12–22.5% [55], while a similar result was found by Walle and Moges (2017) for legume cereal blended with complementary porridge powder [56]. WAI is an indirect measure of the degree of instantaneous reconstitutability of cereals or powders in excess water [57]. No significant differences for the WAI were found among the three instant porridge powders (3.77–4.04 g/g); however, the WAI increased when the protein content increased. Kinsella (1976) explained that the polar amino acid residues of protein bind to water molecules, resulting in a higher water absorption index [58].

Bulk density (BD) represents the structural change of a material. All the instant porridge powders exhibited low BD (0.72 g/cm^3^). Similar results were observed for porridge powders supplemented with soybean and mung bean, with a BD ranging from 0.57 to 0.70 g/cm^3^ [59], and yellow maize porridge, with a BD ranging from 0.69 to 0.81 g/cm^3^ [60]. Low BD is advantageous in the formulation of food for our target demographic, namely school children, because a minimal quantity of powder is necessary to achieve the desired bulkiness of the food products. Higher-BD foods are suitable for use as thickeners, gels and other high-viscosity food products [60,61].

The texture of the instant porridge powders was determined by consistency and the viscosity index. Consistency relates to the firmness, thickness and viscosity of a liquid or fluid semi-solid product. The consistency analysis revealed significant differences among the three instant porridge powders. The higher the PFMF content, the more work or energy was needed to compress the instant porridge powders due to the higher starch content. The viscosity index is regarded as the extrusion energy or work of adhesion. Higher values mean more resistance is required when pulling out the sample. The viscosity index is highly related to resistance to flow and also to cohesiveness and consistency [62,63]. Results indicate that PCPF reduced porridge viscosity, supporting the observation that ICP had the lowest viscosity index.

Chickpea and foxtail millets contain high amounts of macronutrients and micronutrients; however, anti-nutritional factors were also detected. The anti-nutrients determined in this study include phytates, tannins and total phenols. They can combine with nutrients and reduce nutrient bioavailability. Phytate is a negatively charged structure that can bind with positively charged metal ions such as zinc, iron, magnesium and calcium in the digestive tract to form mineral complexes and reduce their bioavailability, eventually leading to mineral deficiency [64]. Thompson (1993) stated that consumption of a 10–60 mg/g phytate diet over a long period resulted in decreased bioavailability of minerals in monogastric animals [65]. Therefore, the phytate composition of all instant porridge powders might not pose a health hazard. Tannins are phenolic compounds that can precipitate protein and decrease protein digestibility and palatability. Tannins can form an insoluble complex with ferric ion (Fe^3+^) and reduce the bioaccessibility of non-haem iron [66,67], while zinc absorption is not affected by tannin [68]. Our results reveal that the combination of chickpea and foxtail millet reduced tannin and total phenol content; however, supplementation of RCP increased these anti-nutrients due to the proportion of ingredients in the formulations. A high concentration of anti-nutrients has an adverse effect but may exert beneficial health effects at adequate amounts by reducing blood glucose levels and the insulin response to starchy foods and/or plasma cholesterol and triglyceride to reduce cancer risk [69,70].

Bioaccessibility is defined as the fraction of the total amount of a substance that is released from the food matrix during gastrointestinal digestion and is potentially available for being absorbed [71]. IPP showed a higher bioaccessibility and percentage of bioaccessibility of iron due to the higher concentration of organic acids in RCP. Organic acids can improve iron absorption and stabilize its soluble form in the small intestine by changing the state of iron oxidation from ferric ion (Fe^3+^) to ferrous ion (Fe^2+^) [72]. A Caco-2 cell study suggested mechanisms underlying the boosting effect of organic acids on two types of iron [73]. Organic acids can improve ferric iron uptake both by chelation and by decreasing the pH, but ferrous iron uptake is promoted only by lowering the pH. No significant difference in zinc content was found in the three instant porridge powders, but IPP showed the highest bioaccessibility and percentage of zinc bioaccessibility. This result suggests that the increase in the bioaccessibility and percentage of bioaccessibility occurred for iron as well as zinc. Organic acids prevented the production of insoluble zinc–phytate complexes [74], like in an in vitro (dialyzability) study by Van der Merwe et al. (2019), who suggested that adding 5.56% and 15% of RCP enhanced the percentage of bioaccessible iron (by 42% and 55%, respectively) and amount of bioaccessible iron (by 107% and 269%, respectively) [5]. Our study provides valuable information on how to improve the bioaccessibility of iron and zinc using RCP in plant-based foods. However, an intervention study needs to be conducted to confirm the improved status because bioavailability has a physiological endpoint [75].

## 5. Conclusions

To reduce the risk of malnutrition, iron deficiency anemia and zinc deficiency retardment, instant porridge powders were successfully developed using PCPF, PFMF and RCP. The ingredients used resulted in significant changes in the physical properties of the instant porridge powders. Addition of PFMF caused an increase in consistency and the viscosity index. Instant porridge powders (ICP, ICF and IPP) are high in protein, fiber and minerals and provide a reasonable percentage of children’s daily requirements for protein, iron and zinc according to the RDA and AR. Addition of RCP in IPP augmented the bioaccessible content and percentage of bioaccessibility of iron and zinc. Roselle calyx powder could be used as a prominent source of iron and organic acids to improve iron content and bioaccessibility of iron and zinc from locally available but often mineral-inhibitor-rich plant-based food sources. The process of preparing instant porridge powders is suitable on a small-scale basis, and the product is safe for consumption. However, the stability studies on the instant pulse porridge powders under long-term and accelerated conditions were the limitations of this study, which should be further investigated. Additionally, even though in vitro dialyzability has been frequently used as a trustworthy predictor for evaluating mineral bioavailability or the contents of ingested iron or zinc that pass through intestinal cell, in vivo experiments should also be further performed to confirm the amount of iron and zinc being absorbed into our body. Our results demonstrate that the developed instant porridge powders had better nutritional profiles that could help to prevent and treat undernutrition and micronutrient deficiencies in young children living in low-to-middle income countries. Additionally, these instant porridge powders might be a prototype either for school meals or in the food industry for development and introduction of new food poducts on the market.

## Figures and Tables

**Table 1 nutrients-14-04070-t001:** Ingredients of three different instant porridge powder formulations.

Ingredients (%)	Instant Porridge Powders		
	ICP	ICF	IPP
Pregelatinized chickpea flour (PCPF)	82.66	49.74	46.76
Pregelatinized foxtail millet flour (PFMF)	–	33.45	31.44
Oil	8.51	7.23	7.23
Roselle calyx powder (RCP)	–	–	4.99
Seasoning spices mix powder (SSMP)	6.70	7.32	7.32
Salt	2.13	2.26	2.26

ICP: instant chickpea powder using PCPF; ICF: instant composite flour using PCPF and PFMF; IPP: instance pulse porridge power using PCPF, PFMF and RCP.

**Table 2 nutrients-14-04070-t002:** Proximate, iron and zinc contents of instant porridge powders (per 100 g dry weight).

Formulations	Energy (kcal)	Protein (g)	Fat (g)	Carbohydrates (g)	Total DF (g)	Ash (g)	Iron (mg)	Zinc ^ns^ (mg)
ICP	458.76 ± 0.44 ^a^	20.15 ± 0.07 ^a^	15.03 ± 0.13 ^a^	60.72 ± 0.25 ^c^	17.15 ± 0.53 ^a^	4.10 ± 0.05 ^b^	4.15 ± 0.01 ^b^	3.58 ± 1.09
ICF	447.07 ± 0.89 ^c^	16.60 ± 0.11 ^b^	12.67 ± 0.23 ^c^	66.66 ± 0.20 ^a^	11.54 ± 0.24 ^c^	4.07 ± 0.08 ^b^	4.23 ± 0.16 ^b^	2.72 ± 0.13
IPP	450.70 ± 0.79 ^b^	16.50 ± 0.00 ^b^	13.73 ± 0.00 ^b^	65.40 ± 0.07 ^b^	13.80 ± 0.00 ^b^	4.44 ± 0.01 ^a^	4.89 ± 0.25 ^a^	3.95 ± 0.21

All data are shown as the mean ± standard deviation (SD) of triplicate determination (*n* = 3). Different lowercase letters denote significantly different contents of the same proximate composition or iron content at *p* < 0.05, while ^ns^ denotes no significant differences in zinc content at *p* ≥ 0.05 in different instant porridge powder formulations using one-way ANOVA, followed by Duncan’s multiple comparison test. DF: dietary fiber; ICP: instant chickpea powder using pregelatinized chickpea flour (PCPF); ICF: instant composite flour using PCPF and pregelatinized foxtail millet flour (PFMF); IPP: instant pulse porridge powder using PCPF, PFMF and roselle calyx powder (RCP).

**Table 3 nutrients-14-04070-t003:** Sensory evaluation of instant porridge powders.

Instant Porridge Powders	Sensory Attributes					
	Appearance ^ns^	Color ^ns^	Odor ^ns^	Taste ^ns^	Texture ^ns^	Overall Liking ^ns^
ICP	6.45 ± 1.21	6.85 ± 1.09	7.10 ± 1.66	6.28 ± 1.81	6.20 ± 1.41	6.43 ± 1.69
ICF	6.65 ± 1.37	6.68 ± 1.21	6.68 ± 1.44	6.75 ± 1.11	6.55 ± 1.11	6.68 ± 1.27
IPP	6.73 ± 1.38	6.70 ± 1.36	7.10 ± 1.21	6.60 ± 1.35	6.73 ± 0.93	6.85 ± 1.10

All data are shown as the mean ± standard deviation (SD) of 40 untrained panelists (*n* = 40). The ^ns^ denotes no significantly different values at *p* ≥ 0.05 for the same sensory attributes in different instant porridge powder formulations using one-way ANOVA, followed by Duncan’s multiple comparison test. ICP: instant chickpea powder using pregelatinized chickpea flour (PCPF); ICF: instant composite flour using PCPF and pregelatinized foxtail millet flour (PFMF); IPP: instant pulse porridge powder using PCPF, PFMF and roselle calyx powder (RCP).

**Table 4 nutrients-14-04070-t004:** Physical properties of instant porridge powders.

Formulations	Color			A_w_	WSI (%) ^ns^	WAI (g/g) ^ns^	BD (g/cm^3^) ^ns^	Consistency (N s)	Index of Viscosity (N s)
	L*	a*	b*						
ICP	35.40 ± 0.27 ^a^	3.69 ± 0.09 ^b^	35.88 ± 0.09 ^a^	0.17 ± 0.00 ^a^	6.40 ± 3.67	4.04 ± 0.12	0.72 ± 0.01	96.65 ± 4.37 ^c^	2.47 ± 2.10 ^c^
ICF	34.06 ± 0.57 ^b^	3.42 ± 0.17 ^c^	32.47 ± 0.26 ^b^	0.13 ± 0.01 ^b^	6.40 ± 1.39	4.00 ± 0.09	0.72 ± 0.00	413.94 ± 35.38 ^a^	62.07 ± 8.93 ^a^
IPP	27.80 ± 0.40 ^c^	4.13 ± 0.02 ^a^	19.64 ± 0.31 ^c^	0.18 ± 0.01 ^a^	6.13 ± 1.22	3.77 ± 0.17	0.72 ± 0.01	306.64 ± 22.65 ^b^	37.65 ± 5.18 ^b^

All data are shown as the mean ± standard deviation (SD) of triplicate determination (*n* = 3). Different lowercase letters denote significant differences in color, A_w_ or texture at *p* < 0.05, while ^ns^ denotes no significant differences in WSI, WAI or BD at *p* ≥ 0.05 in different instant porridge powder formulations using one-way ANOVA, followed by Duncan’s multiple comparison test. ICP: instant chickpea powder using pregelatinized chickpea flour (PCPF); ICF: instant composite flour using PCPF and pregelatinized foxtail millet flour (PFMF); IPP: instant pulse porridge powder using PCPF, PFMF and roselle calyx powder (RCP); A_w_: water activity; WSI: water solubility index; WAI: water absorption index; BD: bulk density; N s: Newton second; L*: lightness; a*: red/green value; b*: blue/yellow value.

**Table 5 nutrients-14-04070-t005:** Mineral inhibitor content of instant porridge powders.

Formulation	Mineral Inhibitors		
	Phytic Acid (g/100 g DW) ^ns^	Tannin (mg TAE/100 mg DW)	Total Phenols (mg TAE/100 mg DW)
ICP	1.48 ± 0.05	1.39 ± 0.01 ^b^	1.67 ± 0.01 ^b^
ICF	1.45 ± 0.13	0.71 ± 0.01 ^c^	1.15 ± 0.02 ^c^
IPP	1.58 ± 0.16	2.58 ± 0.13 ^a^	2.83 ± 0.13 ^a^

All data are shown as the mean ± standard deviation (SD) of triplicate determination (*n* = 3). Different lowercase letters denote significantly different values of tannin or total phenol at *p* < 0.05, while ^ns^ denotes no significant differences in phytic acid content at *p* ≥ 0.05 in different instant porridge powder formulations using one-way ANOVA, followed by Duncan’s multiple comparison test. ICP: instant chickpea powder using pregelatinized chickpea flour (PCPF); ICF: instant composite flour using PCPF and pregelatinized foxtail millet flour (PFMF); IPP: instant pulse porridge powder with PCPF, PFMF and roselle calyx powder (RCP); TAE: tannin acid equivalent; DW: dry weight.

**Table 6 nutrients-14-04070-t006:** In vitro iron and zinc bioaccessibility and pH of instant porridge powders.

Formulations	Iron		Zinc		pH
	Bioaccessible (mg/100 g)	% Bioaccessibility	Bioaccessible (mg/100 g)	% Bioaccessibility	
ICP	0.32 ± 0.01 ^b^	7.83 ± 0.08 ^b^	0.87 ± 0.01 ^b^	30.97 ± 2.38 ^b^	5.78 ± 0.02 ^a^
ICF	0.33 ± 0.02 ^b^	8.20 ± 0.07 ^b^	0.90 ± 0.02 ^b^	34.65 ± 2.66 ^b^	5.83 ± 0.02 ^a^
IPP	0.51 ± 0.01 ^a^	11.08 ± 0.86 ^a^	1.70 ± 0.02 ^a^	45.83 ± 2.90 ^a^	4.55 ± 0.03 ^b^

All data are shown as the mean ± standard deviation (SD) of triplicate determination (*n* = 3). Different lowercase letters denote significantly different values of in vitro iron and zinc bioaccessibility and pH at *p* < 0.05 in different instant porridge powder formulations using one-way ANOVA, followed by Duncan’s multiple comparison test. ICP: instant chickpea powder using pregelatinized chickpea flour (PCPF); ICF: instant composite flour using PCPF and pregelatinized foxtail millet flour (PFMF); IPP: instant pulse porridge powder using PCPF, PFMF and roselle calyx powder (RCP).

**Table 7 nutrients-14-04070-t007:** Microbiological quality assessment of instant porridge powders.

Formulations	Microbiological Quality	
	Total Plate Count (CFU/g)	Yeast and Mold (CFU/g)
ICP	3.0 × 10^4^ ± 0.06	Less than 10
ICF	7.2 × 10^4^ ± 0.03	Less than 10
IPP	6.8 × 10^3^ ± 0.01	Less than 10

All data are shown as the mean ± standard deviation (SD) of triplicate determination (*n* = 3). ICP: instant chickpea powder using pregelatinized chickpea flour (PCPF); ICF: instant composite flour using PCPF and pregelatinized foxtail millet flour (PFMF); IPP: instant pulse porridge powder using PCPF, PFMF and roselle calyx powder (RCP).

**Table 8 nutrients-14-04070-t008:** Recommended dietary allowances (RDAs) and absolute requirement (AR) of protein, iron and zinc of instant porridge powders and their contribution to RDA and AR.

Formulations	Minerals	RDA ^1^	Contribution (%) per Serving (70 g)	AR (mg/d)	Contribution (%) per Serving (70 g)
ICP	Protein	23	58.89	NA	NA
	Iron	15	19.04	0.89 ^2^	25.17 ^2^
	Zinc	5.90	33.46	0.83 ^3^ (1.53 ^4^)	73.38 ^3^ (39.87 ^4^)
ICF	Protein	23	48.51	NA	NA
	Iron	15	18.99	0.89 ^2^	25.96 ^2^
	Zinc	5.90	30.97	0.83 ^3^ (1.53 ^4^)	75.90 ^3^ (41.18 ^4^)
IPP	Protein	23	47.17	NA	NA
	Iron	15	21.40	0.89 ^2^	40.10 ^2^
	Zinc	5.90	44.02	0.83 ^3^ (1.53 ^4^)	143.40 ^3^ (77.80 ^4^)

^1^ Recommended dietary allowances of protein, iron and zinc for children aged 7–9 years are according to Indian Council of Medical Research (ICMR) 2020; ^2^ Absolute requirement of iron (requirement for growth + basal losses + menstrual losses) was based on WHO/UNICEF/UNU 2004 (95th percentile) for children aged 7–10 years; ^3^ Absolute requirement of zinc for children aged 4–8 years with reference body weight of 21 kg as according to IZiNCG (2004); ^4^ Absolute requirement of zinc for children aged 9–13 years with reference body weight of 38 kg as according to IZiNCG (2004); ICP: instant chickpea powder using pregelatinized chickpea flour (PCPF); ICF: instant composite flour using PCPF and pregelatinized foxtail millet flour (PFMF); IPP: instant pulse porridge powder using PCPF, PFMF and roselle calyx powder (RCP); RDA: recommended dietary allowances; AR: absolute requirement; NA: not available.

## Data Availability

Data are contained within this article.

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
