# Peer review of "Food Fortification of Instant Pulse Porridge Powder with Improved Iron and Zinc Bioaccessibility Using Roselle Calyx"

_nutrients, 2022, doi:10.3390/nu14194070_

Round 1

Reviewer 1 Report

1.      Page 13, Line 478. The authors concluded that Roselle calyx powder could improve iron content and bioavailability of iron and zinc. Did the authors evaluate the optimal ratio of the Roselle calyx in the IPP formulation?

2.      I would suggest the authors perform stability studies, including long-term and accelerated conditions.

Reviewer 2 Report

Comment for authors

The manuscript “Food Fortification of Instant Pulse Porridge Powder with Improved Iron and Zinc Bioaccessibility using Roselle Calyx” revealed an overview highlighting the development of nutrient-dense instant porridge powder with pregelatinized chickpea flour (PCPF) and pregelatinized foxtail millet and Bioaccessibility of Iron. The authors expressed and summarized all aspects of but still needed medication for better insight.

* Abstract needs to be rewritten. It should be concise, and simple and highlight the mechanism, and problems, along with the novelty statement which will be confirmed by the results for better insight.

*Abbreviate ICP, ICF, and IPP in the abstract.

*There are some typos and grammatical errors that need to be addressed properly. Present the consistency. Authors are strongly suggested to seek professional help. The expression of many sentences is too long to understand in a manuscript. Concise them into smaller ones. (introduction contains mostly). Mostly need to be rewritten and have typos errors as one example

* did the author check the availability of the zinc and iron?

*How about the thermal stability of the product and its impact on the nutrient content? Moreover, the impact of storage on the color parameters.

*There is a need for animal study (in vivo).

*Inconsistency regarding the format of digits and units (spacing format consistency)

*How does the author compare the novelty and comparison statement from the previously published and its industrial application?

*How about the practical application and benefit to which community?

*There must be a schematic diagram for a clear and better understanding for the reader.
